# Parallel Training of Deep Networks with Local Updates

## Abstract

Deep learning models trained on large data sets have been widely successful in both vision and language domains. As state-of-the-art deep learning architectures have continued to grow in parameter count so have the compute budgets and times required to train them, increasing the need for compute-efficient methods that parallelize training. Two common approaches to parallelize the training of deep networks have been data and model parallelism. While useful, data and model parallelism suffer from diminishing returns in terms of compute efficiency for large batch sizes. In this paper, we investigate how to continue scaling compute efficiently beyond the point of diminishing returns for large batches through *local parallelism*, a framework which parallelizes training of individual layers in deep networks by replacing global backpropagation with truncated layer-wise backpropagation. Local parallelism enables fully asynchronous layer-wise parallelism with a low memory footprint, and requires little communication overhead compared with model parallelism. We show results in both vision and language domains across a diverse set of architectures, and find that local parallelism is particularly effective in the high-compute regime.

## 1 Introduction

Backpropagation (Rumelhart et al., 1985) is by far the most common method used to train neural networks. Alternatives to backpropagation are typically used only when backpropagation is impractical due to a non-differentiable loss (Schulman et al., 2015), non-smooth loss landscape (Metz et al., 2019), or due to memory and/or compute requirements (Ororbia et al., 2020). However, progress in deep learning is producing ever larger models in terms of parameter count and depth, in vision (Hénaff et al., 2019; Chen et al., 2020), language (Radford et al., 2019; Brown et al., 2020), and many other domains (Silver et al., 2017; Vinyals et al., 2019; Berner et al., 2019). As model size increases, backpropagation incurs growing computational, memory, and synchronization overhead (Ben-Nun & Hoefler, 2018). This raises the question of whether there are more efficient training strategies, even for models and losses that are considered well matched to training by backpropagation.

Much of the work on training large scale models focuses on designing compute infrastructure which makes backpropagation more efficient, despite growing model size (Dean et al., 2012b; Chen et al., 2015; Sergeev & Balso, 2018). One of the most common ways to achieve efficient training of deep neural networks with backpropagation is to scale utilizing *data parallelism* (Zhang et al., 1989; Chen et al., 2016), training on bigger batch sizes spread across multiple devices. However, diminishing returns have been reported with this method for larger batch sizes, effectively wasting compute (Goyal et al., 2017; Masters & Luschi, 2018; Shallue et al., 2018; McCandlish et al., 2018). Training based on *pipeline parallelism* has also been introduced, but still requires large batches for efficient training (Petrowski et al., 1993; Ben-Nun & Hoefler, 2018; Huang et al., 2019). Moreover, in addition to the limitation that in the forward pass each layer can only process the input data in sequence (*forward locking*), the use of backpropagation implies that the network parameters of each layer can only be updated in turn after completing the full forward pass (*backward locking*). This backward locking results in increased memory overhead, and precludes efficient parallel processing across layers (Jaderberg et al., 2017). The challenges of scaling compute infrastructure to support deep networks trained with backpropagation motivate the need for alternative approaches to training deep neural networks.

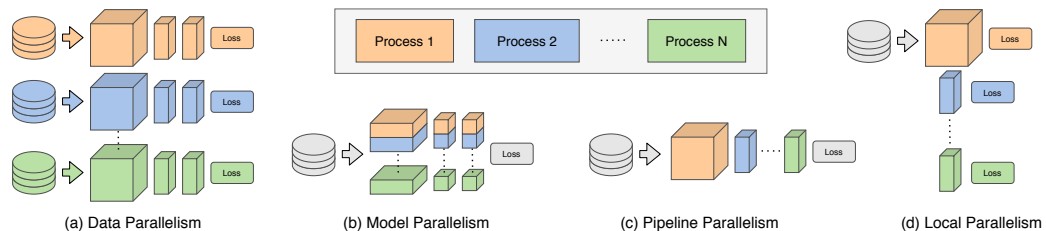

Figure 1: Parallelization in deep learning – (a) data, (b) model, (c) pipeline and (d) local parallelism. While data, model, and pipeline parallelism are existing paradigms for parallelizing learning, we investigate another way of parallelizing learning through local layer-wise training shown in (d).

In this work, we explore how layer-wise local updates (Belilovsky et al., 2019a; Löwe et al., 2019; Xiong et al., 2020) can help overcome these challenges and scale more efficiently with compute than backpropagation. With local updates, each layer is updated before even completing a full forward pass through the network. This remedies the forward and backward locking problems which harm memory efficiency and update latency in standard backprop. Layer-wise local updates are not proportional to gradients of the original loss, and are not even guaranteed to descend a loss function. Nevertheless, in practice they are effective at training neural networks. We refer to this approach of parallelizing compute, which is alternative and complementary to data and model parallelism, as *local parallelism*.

Our investigation focuses on the trade-offs of using local update methods as opposed to global backpropagation. To summarize our contributions: (i) We provide the first large scale investigation into local update methods in both vision and language domains. We find training speedups (as measured by the reduction in required sequential compute steps) of up to $10\times$ on simple MLPs, and $2\times$ on Transformer architectures. These training speedups are the result of local training methods being able to leverage more parallel compute than backprop. (ii) We provide insight into how local parallelism methods work, and experimentally compare the similarity of their gradient and features to those from backprop. (iii) We demonstrate a prototype implementation of local parallelism for ResNets, and show up to a 40% increase in sample throughput (number of training points per second) relative to backprop, due to higher hardware utilization. We believe that local parallelism will provide benefits whenever there are diminishing returns from data parallelism, and avoid stale weights from pipelined model parallelism. Additionally, we have released code showing an example of local parallelism, available at `hiddenurl`.

## 2 RELATED WORK

### 2.1 PARALLELIZATION IN DEEP LEARNING

Scaling large models has led to the development of a number of techniques to train deep models in a parallel fashion (Ben-Nun & Hoefler, 2018), summarized in Figure 1.

**Data Parallelism**: Data Parallelism (Zhang et al., 1989) is an attempt to speed up training of a model by splitting the data among multiple identical models and training each model on a shard of the data independently. Data parallelism is effectively training with larger minibatches (Kaplan et al., 2020). This creates issues around the consistency of a model which then needs to be synchronized (Deng et al., 2012; Dean et al., 2012a). There are two main ways to synchronize weights across model copies: (i) *Synchronous optimization*, where data parallel training synchronizes at the end of every minibatch (Das et al., 2016; Chen et al., 2016), with a communication overhead that increases with the number of devices; (ii) *Asynchronous optimization* that implements data parallel training with independent updates of local model parameters without global synchronization (Niu et al., 2011; Dean et al., 2012a) – this increases device utilization, but empirically gradients are computed on stale weights, which results in a poor sample efficiency and thus slower overall training time compared to synchronous optimization.

**Model Parallelism**: Model Parallelism is used when a model is too large to fit in the memory of a single device and is instead spread over multiple processors (Krizhevsky et al., 2012; Shazeer et al., 2018; Harlap et al., 2018; Lepikhin et al., 2020). This is increasingly common as state of the art performance continues to improve with increasing model size (Brown et al., 2020). Model parallelism unfortunately has a few downsides: (i) *High communication costs* – the total training time for larger networks can become dominated by communication costs (Simonyan & Zisserman, 2015), which in the worst case can grow quadratically with the number of devices, and can reach up to 85% of the total training time of a large model such as VGG-16 (Harlap et al., 2018; Simonyan & Zisserman, 2015); (ii) *Device under-utilization* – forward propagation and backward propagation are both synchronous operations, which can result in processor under-utilization in model-parallel systems. This problem becomes worse as we increase the number of layers (Ben-Nun & Hoefler, 2018; Jia et al., 2014; Collobert et al., 2011; Abadi et al., 2016; Huang et al., 2018).

**Pipeline Parallelism**: Due to the forward and backward locking, using multiple devices to process consecutive blocks of the deep model would make an inefficient use of the hardware resources. Pipelining (Harlap et al., 2018) concurrently passes multiple mini-batches to multiple layers on multiple devices. This increases device utilization but can introduce staleness and consistency issues which lead to unstable training. Harlap et al. (2018) alleviates the consistency issue by storing past versions of each layer. Huang et al. (2019) addresses the staleness issue by pipelining microbatches and synchronously updating at the end of each minibatch. Guan et al. (2019) builds on this work by introducing a weight prediction strategy and Yang et al. (2020) investigates to what extent the tradeoff between staleness/consistency and device utilization is necessary. Local updates on the other hand can keep device utilization high with both small and large batches and avoid the weight staleness problem.

**Local Learning Rules**: Local learning describes a family of methods that perform parameter updates based only on local information, where locality is defined as dependence of neighboring neurons, layers, or groups of layers. The earliest local method we are aware of is Hebbian Learning (Hebb, 1949) which has further been explored in BCM theory (Izhikevich & Desai, 2003; Coesmans et al., 2004), Oja's rule (Oja, 1982), Generalized Hebbian Learning (Sanger, 1989), and meta-learned local learning rules (Bengio et al., 1990; 1992; Metz et al., 2018; Gu et al., 2019). Architectures like Hopfield Networks (Hopfield, 1982) and Boltzmann Machines (Ackley et al., 1985) also employ a local update, and predate backprogation in deep learning. Modern variants of local training methods have attempted to bridge the performance gap with backpropagation. These include projection methods such as Hebbian learning rules for deep networks (Krotov & Hopfield, 2019; Grinberg et al., 2019; Ryali et al., 2020), and local layer-wise learning with auxiliary losses (Belilovsky et al., 2019a;b). Most similar to our work is decoupled greedy layer-wise learning (Belilovsky et al., 2019b; Löwe et al., 2019), which trained auxiliary image classifiers greedily, and local contrastive learning (Xiong et al., 2020). These methods mainly focus on matching the performance of backpropagation with respect to training epochs, whereas our work focuses on tradeoffs. Finally, while not local in the sense that parallelized layers still optimize for the global objective, Huo et al. (2018b) parallelize layers by caching gradients and using delayed gradient signals to overcome the backward locking problem and update decoupled layers in parallel.

## 3 LOCAL PARALLELISM

Given a deep neural network, we divide the layers into a sequence of $J$ blocks, which may contain one or more layers. Each block is trained independently with an auxiliary objective, and receives the activations output by the previous block as input or, in the case of the first block, the data from the sampled minibatch. We consider five variants to train this sequence of $J$ blocks: backpropagation, greedy local parallelism, overlapping local parallelism, and chunked local parallelism, as shown in Figure 2. We also include a baseline method of just training the last, or last two, layers. In all of the local methods, training occurs by attaching objective functions to the end of each block and back propagating the signal locally into the corresponding block or blocks. In this work the auxiliary objective functions that we use take the same form as the global objective. For example, to train a classifier on CIFAR-10, we attach auxiliary linear classifiers to each local block. See Belilovsky et al. (2019b) for further discussion on the form of this objective.

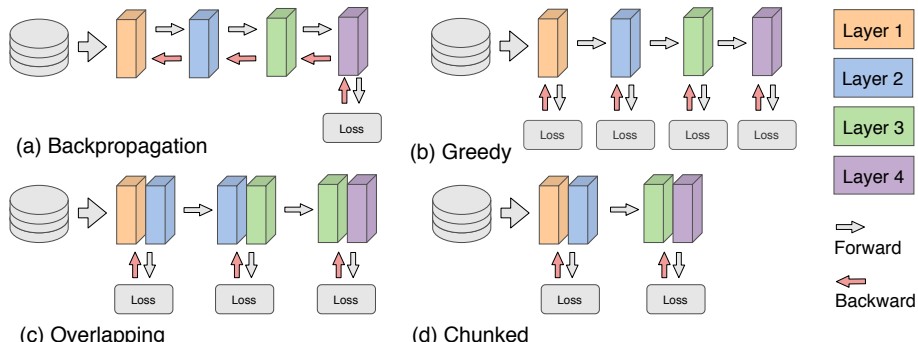

Figure 2: A comparison forward progagation and backward propagation patterns for the architectures considered in this work – (a) backpropagation, (b) greedy local updates, (c) overlapping local updates, and (d) chunked local updates.

*Backpropagation:* In our notation, backpropagation groups all layers into one block and thus $J = 1$. The parameters are updated with one instance of global error correction. While backpropagation ensures that all weights are updated according to the final output loss, it also suffers from forward and backward locking (Jaderberg et al., 2017), an issue that local parallelized methods aim to resolve.

*Greedy local parallelism:* A straightforward approach to enable local training is to attach an auxiliary network to each local layer, which generates predictions from the activations of hidden layers. After generating predictions, each local gradient is backpropagated to its respective local block, shown in Figure 2(b). The activations are then passed as input to the next layer. We refer to this approach, introduced in (Belilovsky et al., 2019b), as *greedy*. Greedy local parallelism is the most parallelizable of all the schemes we consider. However, a potential downside is that fully greedy updates force the layers to learn features that are only relevant to their local objective and preclude inter-layer communication, which may result in lower evaluation performance for the global objective, or worse generalization.

*Overlapping local parallelism:* One issue with the purely greedy approach is that features learned for any individual block may not be useful for subsequent blocks, since there is no inter-block propagation of gradient. For this reason, we consider *overlapping* local architectures where the first layer of each block is also the last layer of the previous block, as shown in Figure 2(c), though overlapping of more layers is also possible. This redundancy enables inter-block propagation of gradient that is still local, since only neighboring blocks overlap. However, this comes at the cost of running additional backward passes. The overlapping architecture has appeared before in Xiong et al. (2020), but was used only for contrastive losses. Ours is the first work to investigate overlapping local architectures for standard prediction objectives in computer vision and language. Overlapping updates are parallelizable, but come with the additional complexity of keeping duplicates of the overlapping components and averaging updates for these layers.

*Chunked local parallelism*: The greedy architecture is maximally parallel in the sense that it distributes one layer per block. However, it is also possible to have fewer parallel blocks by combining multiple layers into one. We refer to this architecture, shown in Figure 2(d), as *chunked* local parallelism. This method trades off parallelizability and therefore throughput for an error signal that propagates through more consecutive layers. It differs from overlapping local parallelism by not needing to duplicate any layer. While previous work has investigated the asymptotic performance of chunked parallelism (Belilovsky et al., 2019b), ours is the first to consider the compute efficiency and parallelizability of local parallelism. By stacking multiple layers per each parallelized block, chunked parallelism sits between fully parallelized methods, such as greedy and overlapping updates, and fully sequential methods like backpropagation.

## 4    EFFICIENT TRAINING ON PARETO FRONTIERS

We explore the trade off between total computational cost and the amount of wallclock time needed to train a particular machine learning model to a target performance, similar to the analysis in Mc-

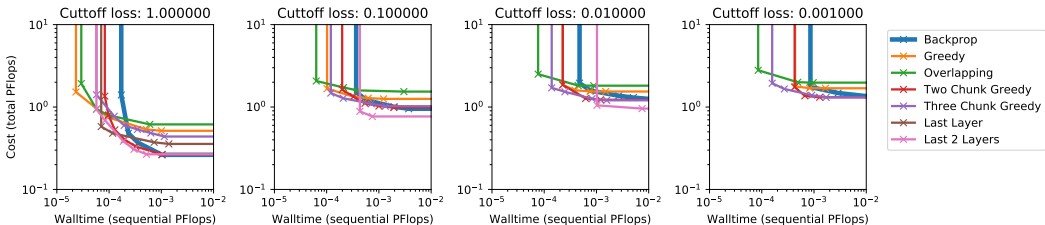

Figure 3: Pareto optimal curves showing the cost vs time tradeoff for an 8-layer, 4096 unit MLP trained on CIFAR-10 reaching a particular cutoff in training loss. We find that under no circumstance is backprop the most efficient method for training. '×' symbol denotes trained models.

Candlish et al. (2018). We use floating point operations (FLOPs) as our unit of both cost and time, as they do not couple us to a particular choice of hardware. Cost is proportional to the total FLOPs used. We report time as the number of sequential FLOPs needed assuming we can run each example, and in the case of the local methods, each layer, in parallel. We refer the reader to Appendix A for detailed information on how total and sequential FLOPs are computed for each experiment.

We compare how backpropagation scales with compute across a variety of local methods: (i) greedy (Figure 2(b)), (ii) overlapping (Figure 2(c)), (iii) two and three chunk greedy (Figure 2(d)), where we split the network into two or three pieces that are trained in a greedy fashion, (iv) last layer & last two layers, a simple baseline where we only backpropagate through the last one or two layers and keep the rest of the network parameters fixed. We apply these methods on a variety of architectures and data including a dense feed-forward network, a ResNet50 network (He et al., 2016) trained on ImageNet (Russakovsky et al., 2015), and a Transformer (Vaswani et al., 2017) model trained on LM1B (Chelba et al., 2013). In Appendix C, we provide results for additional feed-forward networks, a ResNet18 trained on ImageNet, and a larger Transformer, as well as further architecture details. For each model and training method, we perform a large sweep over batch size as well as other optimization hyperparameters, and only display the best-performing runs on the Pareto optimal frontier. See Appendix B for more detail.

The resulting figures all follow the same general structure. Models train with low total cost when the amount of available compute is large. By increasing batch size, the amount of compute utilized per parallel process can be reduced efficiently until a critical batch size is reached, at which point further increasing the batch size results in diminishing returns in terms of compute efficiency, which is similar to results reported for backpropagation in (McCandlish et al., 2018). We find that, in most cases, local updates significantly increase the training speed over deep networks in the high-compute regime, and therefore utilize less total compute than backpropagation. When applicable, we additionally show tables of the best achieved results across all parameters ignoring the time to reach these values. In this setting, we find that backpropagation usually achieves the best performance. This is partially due to the fact that all of these models are trained for a fixed number of examples, and partially due to the fact that backpropagation makes higher use of the capacity of a given model, which we further investigate in Section 5.

## 4.1 Synthetic: MLP's Over-fitting to CIFAR-10

As a proof of concept we first demonstrate optimization performance on an eight layer MLP with 4096 hidden units, performing classification on the CIFAR-10 dataset (Krizhevsky et al., 2009). Hyperparameter and optimization details can be found in Appendix B.1. From the resulting Pareto frontiers shown in Figure 3, we find that in no circumstance is backpropagation the best method to use. In the high compute regime, we find that local methods enable training up to $10\times$ faster (e.g. in 0.001 cutoff).

## 4.2 Language Modeling: Transformers on LM1B

Next we explore a small (6M parameter) Transformer (Vaswani et al., 2017) trained on LM1B (Chelba et al., 2013). We build off of an implementation in Flax Developers (2020). Hyperparameters and optimization details can be found in Appendix B.2. We find that, for the higher cutoffs, many of the local methods vastly outperform backpropagation. For the lower cuttofs ($\leq 4.0$), we

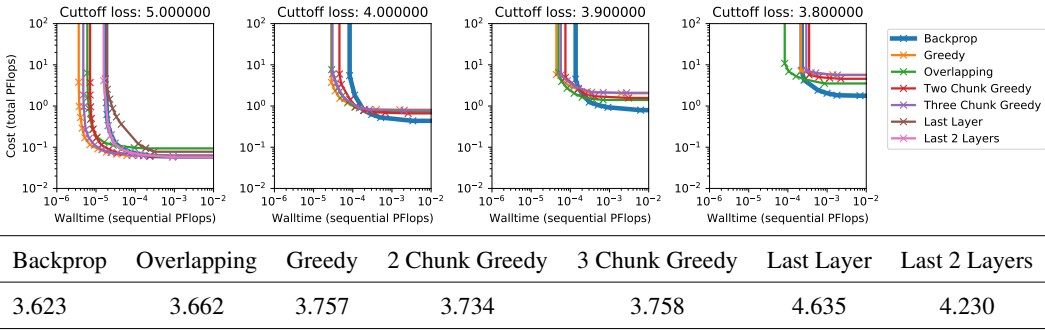

| Backprop | Overlapping | Greedy | 2 Chunk Greedy | 3 Chunk Greedy | Last Layer | Last 2 Layers |
|---|---|---|---|---|---|---|
| 3.623 | 3.662 | 3.757 | 3.734 | 3.758 | 4.635 | 4.230 |

Figure 4: Total compute cost vs. serial compute cost (walltime) Pareto curves computed from validation loss for a 6M parameter parameter transformer. We find that for high loss cutoffs (e.g. 5.), significant speedups (around $4\times$) can be obtained. For cutoffs of 4.0, and 3.9 speedups (around $2\times$) are still possible, but only with the overlapping method. For even lower cut offs, 3.8, we find the majority of our models are unable to obtain this loss. In the bottom table we show the best achieved validation loss for each training method maximized across all hyperparameters.

find that while backpropagation is more efficient in the high-time regime, local methods train significantly faster in the high-compute regime, and can *train $2\times$ faster than backpropagation*. These local methods do not reach as low of a minimum in the given training time however. See Figure 4.

### 4.3 IMAGE CLASSIFICATION: RESNET50 ON IMAGENET

Next we explore performance of optimization parallelism on a ResNet50 model trained on the ImageNet dataset (Russakovsky et al., 2015) (Figure 5). Hyperparameter and configuration details can be found in Appendix C.1. We find, as before, that for many cutoff values local parallelism shows gains over backpropagation in the high-compute regime. However, at the cutoff of 74% these gains shrink and the local methods are slightly less efficient. We hypothesize this is in-part due to increased overfitting by the local methods. To see this we can observe that local methods are much more competitive when evaluated on training accuracy. This suggests that given more data these local methods will be competitive.

### 5 PROBING BEHAVIOR OF LOCAL UPDATES

In the previous section we showed that in some cases local parallelism can provide large speedups over backpropagation but suffers in terms of the best achievable performance. In this section we explore why and how these methods work, and discuss limitations.

**Gradient Angles:** Local parallelism does not follow the gradient of the underlying function. Instead it computes a local, greedy approximation. To check the quality of this approximation we measure the angle between the true gradient, and the gradient computed with our greedy method (Figure 6a). We find positive angles which imply that these directions are still descent directions. As one moves further away from the end of the network these similarities shrink.

**Larger Block Sizes Improve Generalization:** As noted in Huo et al. (2018a;b) and Belilovsky et al. (2019b), using chunked local parallelism with more parallel blocks can decrease performance. Here we show that practically this reduction in performance seems to stem mainly from a worsening generalization gap, with train and test results shown for various chunk sizes in Figure 6. A chunk size of nine is simply backprop, and a chunksize of one is fully greedy.

**Capacity: Ability to Fit Random Labels:** Throughout our work we find that models trained with local updates don't make as efficient use of model capacity. This is not necessarily a problem, but represents a tradeoff. Researchers have found that increased model sizes can be used to train faster without leveraging the extra capacity to its fullest (Raffel et al., 2019; Kaplan et al., 2020). Additionally, techniques like distillation can be used to reduce model size (Hinton et al., 2015). We demonstrate this capacity issue by fitting random labels with a ResNet on CIFAR-10, shown in Figure 6.

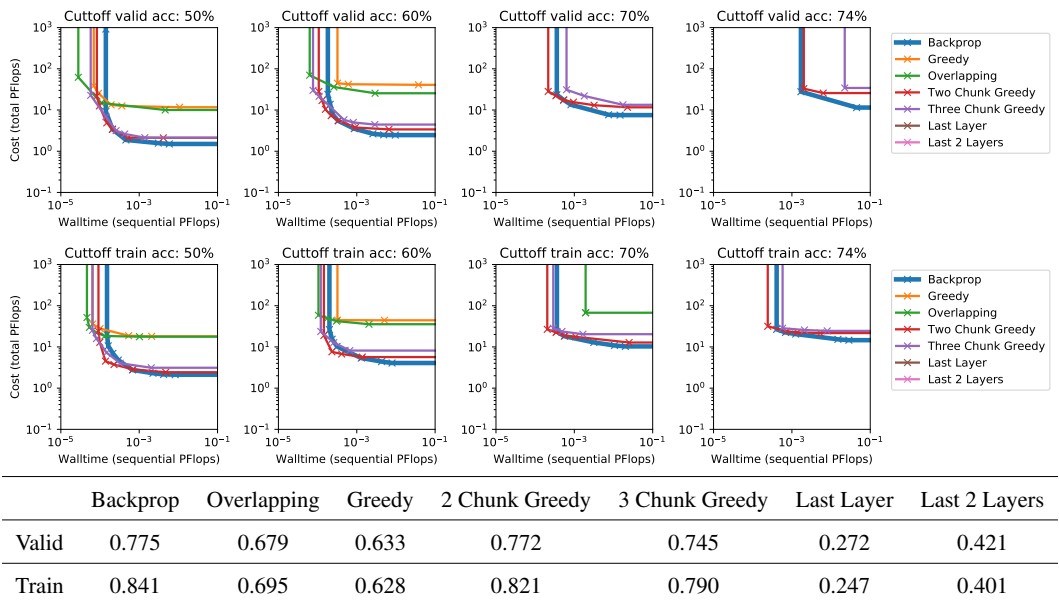

| | Backprop | Overlapping | Greedy | 2 Chunk Greedy | 3 Chunk Greedy | Last Layer | Last 2 Layers |
|---|---|---|---|---|---|---|---|
| Valid | 0.775 | 0.679 | 0.633 | 0.772 | 0.745 | 0.272 | 0.421 |
| Train | 0.841 | 0.695 | 0.628 | 0.821 | 0.790 | 0.247 | 0.401 |

Figure 5: Total compute cost vs walltime frontier for ResNet50 models trained on ImageNet. We show the cost/time to reach a certain cutoff measured on validation accuracy (top) and training accuracy (bottom). With low cutoffs (50%, 60%, 70%), modest speedups can be obtained on validation performance. With higher cutoffs (74%) however backprop is optimal. In the subsequent table, we show the best accuracies reached for each method across all configurations. We find that the least parallelized method, Two Chunk Greedy, is the only local method competitive with backprop on validation accuracy.

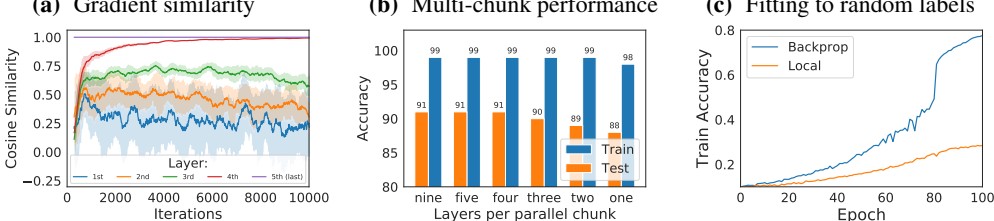

Figure 6: Properties and trade-offs of local parallelism. *(a)* The cosine similarity between backpropagation gradients and greedy local gradients for a 5 layer convolutional neural network. Gradients in the last two layers are identical to, or converge towards, those from backpropagation. Earlier layer local gradients are increasingly dissimilar to those from backpropagation but are still descent directions. *(b)* An ablation of the number of layers per chunk for ResNet18 trained on CIFAR-10. Adding more layers per chunk improves generalization, while the training loss is roughly equal across different chunk sizes. *(c)* Backprop and greedy local training is performed on a ResNet18, trained on CIFAR-10 with random labels. Global backpropagation demonstrates higher capacity, in that it is able to memorize the dataset better than local greedy backpropagation.

**Local Methods Learn Different Features:** One way to show differences between local and non-local methods is to look at the features learned. For each method we test we take the best performing model and visualize the first layer features. The results are shown in Figure 7. Qualitatively, we see similar first layer features from Backprop and Two/Three Chunk local parallelism. The more greedy approaches (Overlap, Greedy) yield a different set of features with fewer edge detectors. Finally, when training with only the last layers, the input layer is not updated, and the features are random.

Backprop Two Chunk Three Chunk Overlap Greedy Last 2

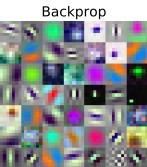 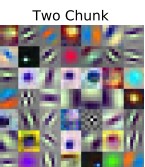 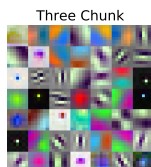 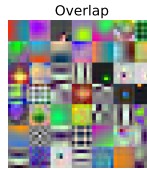 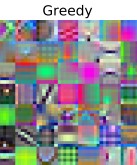 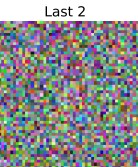

Figure 7: First layer filters taken at the end of training normalized by the min and max value per filter. We find the more global methods (Backprop, Two Chunk, Three Chunk) learn similar distributions over features. However more greedy approaches (Overlap and Greedy) learn visually distinct, less edge-like, features. Finally the Last 2 filters are random, because the input layer is never updated.

## 6 REALIZED PERFORMANCE GAINS

Here we show that performance gains of local parallelism can be realized on real hardware, and that they are similar to or better than pipelined backpropagation despite the increased computation needed for auxiliary losses. We train ResNet34, ResNet50 and ResNet101 (He et al., 2016) on the ImageNet dataset (Deng et al., 2009), and compare throughput (images per second) between chunked local parallelism and synchronous pipelined backpropagation (Huang et al., 2019). We implement the models in TensorFlow (Abadi et al., 2016) and train them across 4 or 8 Intelligence Processing Units (IPUs – see details in Appendix E). Note that neither local nor pipeline configurations make use of data parallelism which could be applied identically in both cases. We use activation recomputation in the case of pipelined backprop (see discussion in Appendix D.3). The results in Table 1 show that chunked local parallelism can achieve similar or greater throughput compared to pipelined backpropagation, for the same local batch size. This provides evidence that local parallelism can enable similar hardware efficiency without necessitating an increase of minibatch size. It is therefore amenable to a greater level of data parallelism before performance degradation due to a large global batch size. The difference in throughput between backpropagation and local parallelism with the same local batch size is primarily due to the poor utilisation during the "ramp-up" and "ramp-down" phases of the pipelined backpropagation. This can be mitigated by running the pipeline in the steady state for more stages (compare rows 4 and 5 of Table 1). However, this results in the accumulation of gradients from a larger number of local batches, thus costing a larger effective batch size. With greedy local parallelism, updates can be applied asynchronously and the pipeline can be run in steady state indefinitely, after an initial ramp-up phase. Hardware utilization analysis and further discussion can be found in Appendix D.

| Network | Local batch size | Backprop batch size | # IPUs | Speedup over backprop |
|---|---|---|---|---|
| ResNet34 | 32 | $32 \times 8$ | 4 | 8% |
| | 32 | $32 \times 16$ | 8 | 37% |
| ResNet50 | 16 | $16 \times 8$ | 4 | 28% |
| | 16 | $16 \times 16$ | 8 | 32% |
| | 16 | $16 \times 32$ | 8 | 12% |
| ResNet101 | 4 | $4 \times 16$ | 8 | 33% |
| | 8 | $8 \times 16$ | 8 | 41% |

Table 1: Increase in throughput for ImageNet training with chunked local updates vs pipelined backprop. Backprop batch size $a \times b$, where $a$ is the microbatch size and $b$ is the number of microbatches over which gradients are accumulated.

## 7 CONCLUSION

In this work we demonstrated that local parallelism is a competitive alternative to backpropagation in the high-compute training regime, and explored design decisions and trade-offs inherent in training with local parallelism. We summarize some main takeaways from our work:

- *Speed vs. Performance:* Greedy local parallelism should be used if speed and compute-efficiency are the primary objectives. Chunked local parallelism should be used if performance is the primary objective.

- *Gains in High-Compute Regime:* Local parallelism can be useful to prolong compute-efficient scaling, and therefore faster training, with larger batch sizes once data parallelism begins to saturate.

- *Comprehensive Analysis:* Local parallelism can be applied across multiple modalities (vision, language) and architectures (MLPs, ResNets, Transformers).

We hope that local methods will enable new research into large models. By lowering communication requirements – particularly latency requirements surrounding synchronization – we believe that local parallelism can be used to scale up and train more massive models in a more distributed fashion.

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

# A  CALCULATION OF TOTAL FLOPS AND SEQUENTIAL FLOPS

To construct the Pareto curves used in this work we need some estimate of compute time. Obtaining hardware independent measurements of compute cost and compute time is desirable, but in general impossible, as different hardware makes different trade offs for compute efficiency. In this work we choose to use a theoretical estimate of compute costs based on floating point operation (FLOP) counting. In all three models, we divide the costs up into three measurements: FLOPs needed for a forward pass through a layer, flops needed for the auxiliary loss computation, and a multiplier to compute the number of flops for a backward pass. For simplicity, we average the compute costs across layers. While this is strictly not feasible in reality with a batch size of one per device, we can come close to approximating it by using more or less parallel hardware per layer. This is relatively simple to implement given the minimal communication overhead. We additionally take into account optimizer flops, which we we approximate as ten times the number of parameters, but this results negligible.

## A.1  CALCULATIONS PER MODEL

**MLP:** An MLP is parameterized by the hidden size, $N$, and the number of layers, $L$. The first layer's total flops are from matrix vector multiplication, a bias add of size $N$, and a ReLU (which we assume costs 1 FLOP per entry). This yields a total size of $(2 * I * N - I) + N + 2 * N$ FLOPs, where $I$ is the input size (Hunger, 2005). The auxiliary classifiers consist of a matrix vector multiplication to size 10, a bias add, and a softmax cross entropy loss. We assume the softmax costs 5 flops per estimate leading to a flop estimate of $(2 * N * 10 - N) + 10 + 5 * 10$. For this problem, we approximate the backward multiplier to be 1.5. For the MLP model used in the main text (with hidden size $N = 4096$ and $L = 8$ layers), the average forward cost per layer is 32514176.0 flops, and the auxiliary loss 77884.0 flops.

For the remaining models, we compute our estimates of these components by first using JAX to convert our models to TensorFlow functions, and then leveraging TensorFlow's *tf.compat.v1.profiler.profiler*.

**ResNet50:** This model has $L = 17$ layers and contains 38711720 parameters. We find that the average forward flop count per example, per layer is 5479411.176470588, the auxiliary loss per layer is 3382457.3529411764, and the backward multiplier is 2.0280375672996596.

**ResNet18:** This model has $L = 9$ layers, and has 13170792 parameters. We find that the average forward flop count per example, per layer is 1640544.352941176, the auxiliary loss flop count per example per layer is 565900.6470588235, and the backward multiplier is 2.08565879129763.

**Transformer small:** This model has $L = 4$ layers. We find that the average forward cost per example, per layer is 13837446.0, the auxiliary loss is 1163904.0, and the backward multiplier is 1.6581083035860107.

**Transformer large:** This model has $L = 6$ layers. We find that the average forward cost per example, per layer is 51037318.0, the auxiliary cost is 4653696.0, and the backward multiplier is 1.7526391044859857.

## A.2  CALCULATIONS PER METHOD

In all cases, we first obtain the total computation cost in terms of flops and then compute time (or sequential flops) by dividing by the max amount of parallelism (assuming that each example and each layer are run concurrently). As stated before, this is not strictly possible to implement in hardware. In reality, however, we expect more than one example to be used per device in combination with data parallelism and thus appropriate load balancing can be done.

All of these calculations are a function of the 4 numbers described above (forward cost, auxiliary cost, backward multiplier and the optimizer cost) in addition to batch size and the number of gradients steps until the target loss is reached.

**Backprop:** Per step, backprop involves running one forward pass and one backward pass of the entire network plus plus one auxiliary head for the last layer loss computation. The cost per example

is computed as follows:

$$\text{cost\_per\_example} = (1 + \text{backward\_multiplier}) * (\text{forward\_cost} * \text{layers} + \text{aux\_cost})$$
$$\text{cost} = \text{cost\_per\_step\_example} * \text{steps} * \text{batch\_size} + \text{steps} * \text{optimizer\_cost}$$
$$\text{time} = \text{cost}/\text{batch\_size}$$

**Greedy:** Per step, the greedy method requires running one forward and backward pass for $L$ layers and $L$ auxilary loss computations.

$$\text{cost\_per\_example} = (1 + \text{backward\_multiplier}) * ((\text{forward\_cost} + \text{aux\_cost}) * \text{layers})$$
$$\text{cost} = \text{cost\_per\_step\_example} * \text{steps} * \text{batch\_size} + \text{steps} * \text{optimizer\_cost}$$
$$\text{time} = \text{cost}/(\text{batch\_size} * \text{layers})$$

**Overlapping:** Because we are using overlapping chunks of layers, additional compute must be performed. This method uses one full forward pass though the entire network plus two backward passes for each non terminal layer. The terminal layer only requires one less layer of computation. We additionally need one forward and backward pass of each auxiliary loss. An additional average of gradients is required which incurs extra compute per layer.

$$\text{cost\_per\_example} = (\text{forward\_cost} + \text{aux\_cost}) * \text{layers} +$$
$$(\text{layers} - 1) * \text{backward\_multiplier} * (2 * \text{forward\_cost} + \text{aux\_cost}) +$$
$$\text{backward\_multiplier} * (\text{forward\_cost} + \text{aux\_cost})$$
$$\text{cost} = \text{cost\_per\_step\_example} * \text{steps} * \text{batch\_size} + \text{steps} * (\text{optimizer\_cost} + 2 * \text{parameters})$$
$$\text{time} = \text{cost}/(\text{batch\_size} * \text{layers})$$

**Two/Three chunk:** In this, we perform a full forward + backward pass for each layer plus two or three auxiliary losses. Lets call the number of chunks $K$ for the equations bellow.

$$\text{cost\_per\_example} = (1 + \text{backward\_multiplier})(\text{forward\_cost} + K * \text{aux\_cost})$$
$$\text{cost} = \text{cost\_per\_step\_example} * \text{steps} * \text{batch\_size} + \text{steps} * \text{optimizer\_cost}$$
$$\text{time} = \text{cost}/(\text{batch\_size} * K)$$

**Last One/Two Layers:** These methods require a full forward pass, a single auxilary loss computation and then a backward pass on the last $K$ layers. To calculate time, we assume this last $K$ layers is the smallest atomic chunk that can be run and we divide up the remaining layers accordingly.

$$\text{cost\_per\_example} = (\text{layers} * \text{forward\_cost} \text{aux\_cost}) + \text{backward\_multiplier} * (K * \text{forward\_cost} + \text{aux\_cost})$$
$$\text{cost} = \text{cost\_per\_step\_example} * \text{steps} * \text{batch\_size} + \text{steps} * \text{optimizer\_cost}$$
$$\text{num\_parallel} = (\text{layers} + K * \text{backward\_mult})/(K * (1 + \text{backward\_mult}))$$
$$\text{time} = \text{cost}/(\text{batch\_size} * \text{num\_parallel})$$

## B    Hyperparameter and Configuration Details for Experimental Results

### B.1    MLP on CIFAR-10

We sweep the batch size from 64-524,288 in powers of 2. At each batch size, we train models using learning rate tuned Adam (with six values log spaced between 1e-4 and 3e-2) as well as the first 50 optimizers taken from opt_list to provide a stronger baseline (Metz et al., 2020). All models are trained for three million examples on an eight core TPU-V2 using gradient accumulation to control memory usage. We select a sequence of cut off values (the loss for which we attempt to reach in the shortest time) and plot the Pareto frontier of the different training methodology in Figure 3.

### B.2    Transformers on LM1B

Our Transformer has 4 layers, 8 heads per attention layer, 128-dimensional query, key, and value vectors, 256-dimensional hidden layers, and 128-dimensional embeddings. We train on length 128 sequences formed from subword tokenization with a vocabulary size of 8k. Each Transformer layer is treated as a separate parallelizable component. Our auxiliary classifiers consist of layer norm and a linear projection back to the vocabulary, with a softmax cross entropy loss. We sweep batch sizes in powers of two from 32 to 524,288. At each batch-size we either train Adam with six different learning rates taken evenly spaced on a log scale between 1e-4 and 3e-2 and the first 50 optimizers from opt_list (Metz et al., 2020). All models are run until they have processed 50 million sequences, an an 8-core TPU-V2 with gradient accumulation to control memory. We chose four cutoff values computed on validation loss to show early in training (a value of 5.0 and 4.0), the value chosen by Shallue et al. (2018) (3.9), and a loss value slightly lower (3.8). Results can be found in Figure 4.

## C    Additional Pareto Curves Experiments

We provide additional Pareto curves for different architecture models.

### C.1    ResNets on ImageNet

We build our code off of the Haiku implementation (Hennigan et al., 2020). We break the network up by putting the first convolution, and each residual block into a separate parallelizable component For auxiliary losses we apply batch normalization (Ioffe & Szegedy, 2015), then ReLU, then compute mean across the spatial dimensions, and finally perform a linear projection to the output classes. We sweep batch sizes from 8 to 524,288 in powers of 2. For each batch size we randomly sample optimizer hyperparameters for both the SGDM optimizer with a staircase schedule described in Goyal et al. (2017) and from the first 50 configurations in opt_list. The resulting cost wall time Pareto curves for both validation accuracy and training accuracy are shown in Figure 5.

### C.2    MLPs

We provide MLP's trained matching Section 4.1 but using a different number of hidden units. In addition to 4096 units, we show 1024, 256, and 64 units in Figure 8. We find the last 2 layers performs well for larger networks, as there is enough capacity, but is considerably less useful as model size shrinks.

### C.3    Transformer Large

In this section we explore a larger transformer than that in Section 4.2. This transformer matches the default settings of of (Flax Developers, 2020). It has has 6 layers, 8 heads per attention layer, 512-dimensional query, key, and value vectors, 512-dimensional hidden layers, and 512-dimensional embeddings. We train on length 128 sequences formed from subword tokenization with a vocab size of 32k. We show results in Figure 9. Unlike in the small transformer and due to increased compute costs, we random sample configurations instead of running all of them.

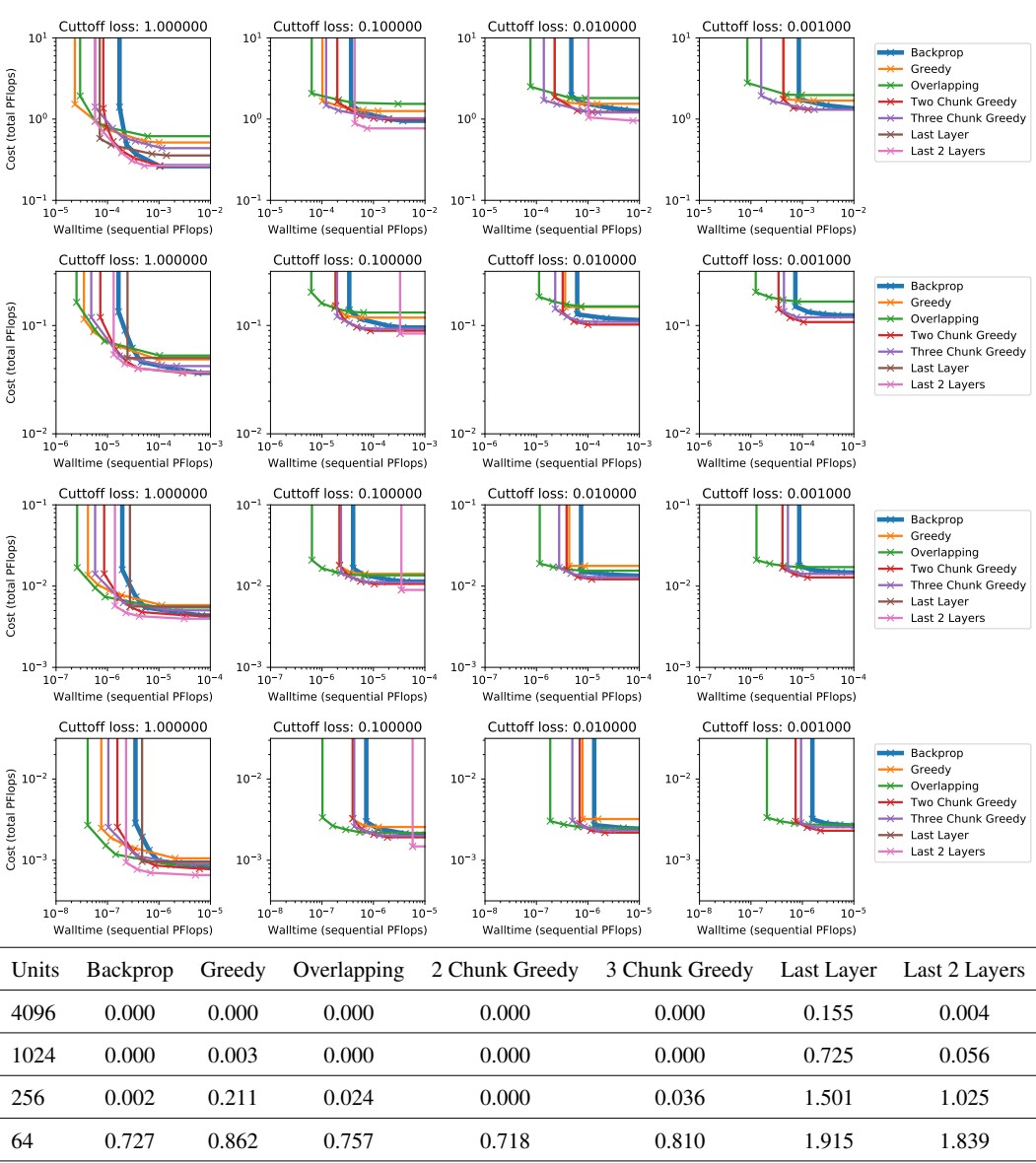

| Units | Backprop | Greedy | Overlapping | 2 Chunk Greedy | 3 Chunk Greedy | Last Layer | Last 2 Layers |
|-------|----------|--------|-------------|----------------|----------------|------------|---------------|
| 4096 | 0.000 | 0.000 | 0.000 | 0.000 | 0.000 | 0.155 | 0.004 |
| 1024 | 0.000 | 0.003 | 0.000 | 0.000 | 0.000 | 0.725 | 0.056 |
| 256 | 0.002 | 0.211 | 0.024 | 0.000 | 0.036 | 1.501 | 1.025 |
| 64 | 0.727 | 0.862 | 0.757 | 0.718 | 0.810 | 1.915 | 1.839 |

Figure 8: Pareto optimal curves showing the cost vs time tradeoff for an 8-layer MLP trained on CIFAR-10 with different number of units. From top to bottom we show 4096, 1024, 256, and 64 hidden unit MLPs. We continue find that under no circumstance is backprop the most efficient method for training. $\times$ denote trained models. In the following table we show the best performance achieved for each different model. We find large models are able to near perfectly minimize this loss. For smaller models we find Backprop, achieves the lowest loss followed by Two Chunk Greedy, then Overlapping.

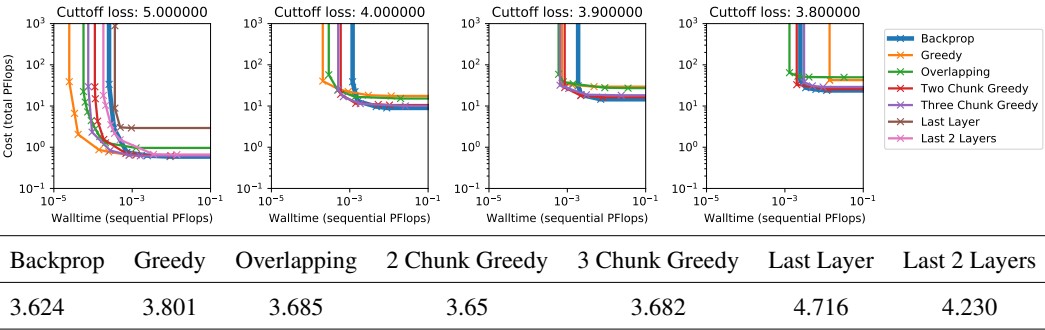

| Backprop | Greedy | Overlapping | 2 Chunk Greedy | 3 Chunk Greedy | Last Layer | Last 2 Layers |
|----------|--------|-------------|----------------|----------------|------------|---------------|
| 3.624 | 3.801 | 3.685 | 3.65 | 3.682 | 4.716 | 4.230 |

Figure 9: Cost wallclock trade off curves for a larger transformer model. We find for high loss cutoffs (e.g. 5.), significant speedups (around $4\times$) can be obtained. For cutoffs of 4.0, and 3.9 speedups (around $2\times$) are still possible, but only with the overlapping method. For even lower cut offs, 3.8, we find the majority of our models are unable to obtain this loss. In the subsequent table we show the best archived validation loss maximized across all configurations.

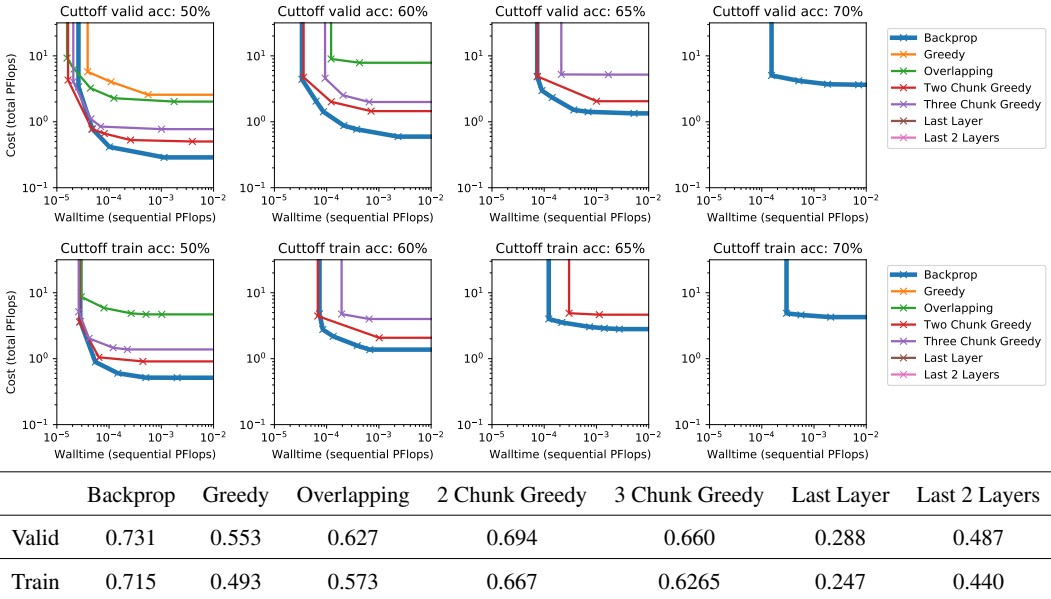

| | Backprop | Greedy | Overlapping | 2 Chunk Greedy | 3 Chunk Greedy | Last Layer | Last 2 Layers |
|-------|----------|--------|-------------|----------------|----------------|------------|---------------|
| Valid | 0.731 | 0.553 | 0.627 | 0.694 | 0.660 | 0.288 | 0.487 |
| Train | 0.715 | 0.493 | 0.573 | 0.667 | 0.6265 | 0.247 | 0.440 |

Figure 10: Cost wallclock time frontier for ResNet18 models trained on ImageNet. We show the cost/time to reach a certain cutoff measured on validation accuracy (top) and training accuracy (bottom). We find with low cutoffs modest speedups can be obtained on validation performance. In the table we report the best achieved accuracy over all hyperparameters.

### C.4 RESNET18

In addition to the ResNet50 we explored in the main text (Section 4.3) we also explore a ResNet18 trained with the same protocols. We find similar results in Figure 10.

## D HARDWARE UTILIZATION

### D.1 EXECUTION TRACES

The execution traces shown in Figure 11 illustrate hardware utilization. A single cycle of pipelined backpropagation over 8 microbatches is shown in Figure 11a. This comprises a ramp-up phase (blue dashed box), a steady state where all processors are used at each step (e.g. green dashed box), and a ramp-down phase (orange dashed box). Gradients for all steps are accumulated and applied at

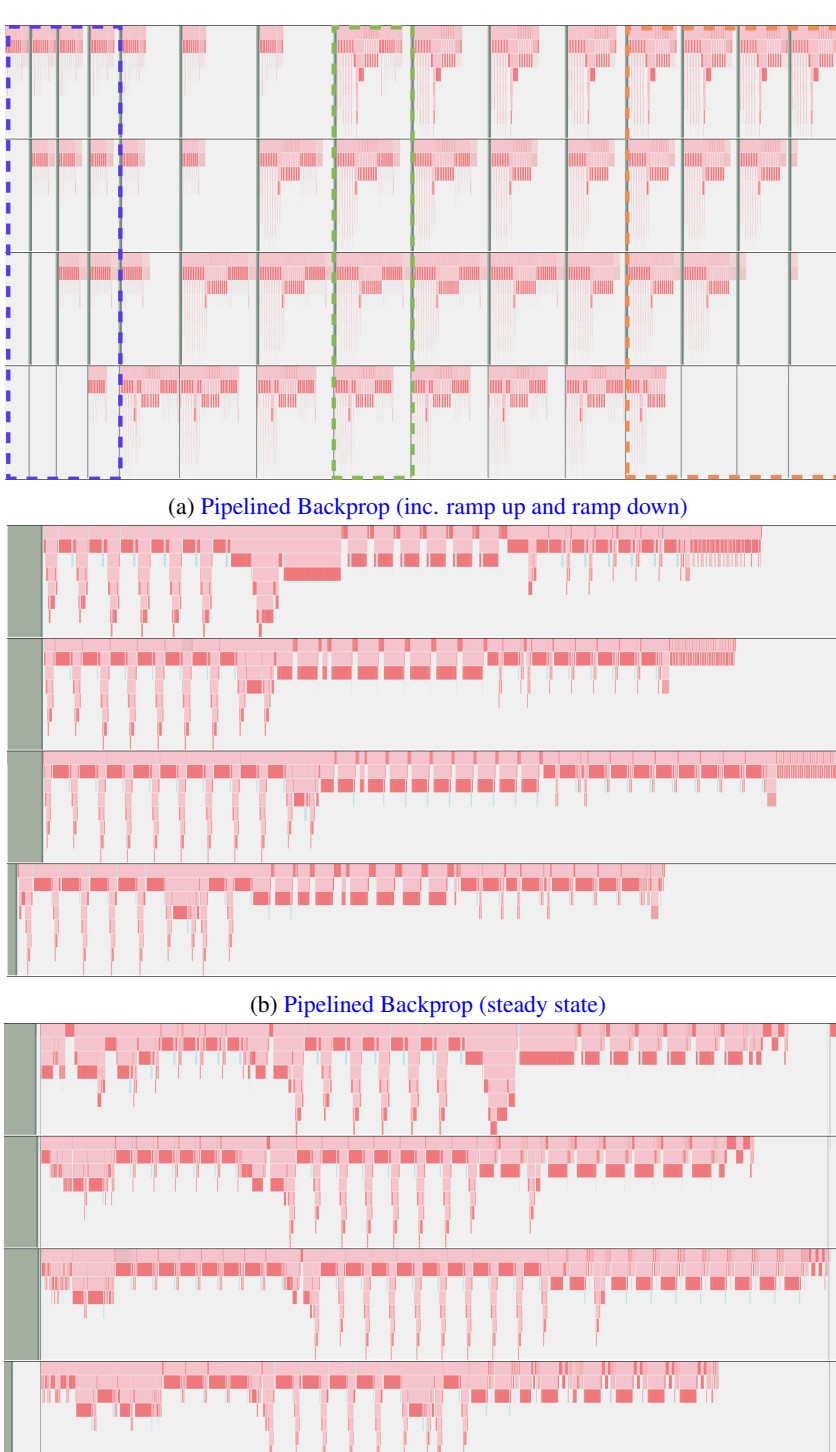

(a) Pipelined Backprop (inc. ramp up and ramp down)

(b) Pipelined Backprop (steady state)

(c) Chunked Local Parallelism

Figure 11: Execution traces for ResNet34 on ImageNet, batch size $8 \times 8$ for backprop and 8 for chunked local parallelism. Green blocks denote inter-IPU communication, pink and red indicates computation and light blue represents exchange of data between tiles on the same IPU. Each swim lane represents the execution trace for one of four IPUs, within which there exist a hierarchy of operations spread vertically within the lane, with the lowest-level operation in the hierarchy highlighted in darker pink. The horizontal axis is hardware cycles. The dashed blue and orange boxes in (a) cover the ramp-up and ramp-down phases of pipelined backprop respectively, and a steady state step is highlighted with the green dashed box and expanded in (b).

| | Chunked Local | | Backprop | |
|---|---|---|---|---|
| IPU | Received | Transmitted | Received | Transmitted |
| 1 | 0 | 12.3 | 12.3 | 12.3 |
| 2 | 12.3 | 6.2 | 18.5 | 18.5 |
| 3 | 6.2 | 3.1 | 9.3 | 9.3 |
| 4 | 3.1 | 0 | 3.1 | 3.1 |

Table 2: Total data communicated between IPUs, for a single step of ResNet34 training on ImageNet, over 4 IPUs. Results are for batch size $32 \times 8$ for pipelined backprop and 32 for chunked local parallelism. All measurements in MB.

the end of the cycle. It is clear that there is significant device underutilization in the ramp-up and ramp-down phases. Moreover, in the steps which follow the ramp-up the gradient signal has not yet reached the earlier layers of the network, preventing backwards passes from being executed on these processors and causing poor load balancing. A similar effect is present in the steps which precede ramp down, where no further forward passes are run in shallower layers of the network. These effects contribute to the higher throughput of local parallelism relative to backpropagation.

During the steps where all processors calculate a forward and backward pass (dashed green box in Figure 11a, enlarged in Figure 11b), the utilization of backpropagation is at its highest. More of these steps may be executed in each cycle, which reduces the fractional overhead of ramp-up and ramp-down. However, this results in gradients being accumulated over more microbatches, and therefore in the use of a larger overall minibatch size, thus reducing the potential for data-parallel replication of the pipeline.

During the peak-utilization steps, the operations executed are largely the same as that for chunked local parallelism (Figure 11c), with the exception of the auxiliary classifiers necessary for local parallelism. Note that in the pipelined backpropagation case (Figure 11b) the backward pass is executed before the forward pass (with different microbatches), whereas in the chunked local update case the forward pass is executed before the backward pass (for the same minibatch). This difference in ordering can be seen in the tall, thin "spike-like" operations which occur at the start of the backwards pass. In the pipelined backpropagation step they are run at the start of the step, while they are run mid way through the local update step.

## D.2 INTER-PROCESSOR COMMUNICATION

Our profiling of the code found that the total data communicated between processors for local parallel training was half that of pipelined backprop, for all network/batch size configurations presented in Table 1. As an example, Table 2 reports the data communicated between IPUs for a single training step of ResNet34. Note this corresponds to a single microbatch in the case of pipelined backpropagation. There is a clear reduction in the total data communicated with local parallelism relative to backpropagation. The reduction is as expected: IPU 1, processing the first layers, receives no data from other IPUs as no gradient signal from later layers are communicated backward. IPU 4 does not transmit any data backward for the same reason. Overall the total data communicated is 43.2MB for chunked local parallelism and 86.4MB for backpropagation. These results are consistent with the expectation that pipelined local parallelism should result in half as much inter-processor communication as pipelined backpropagation.

## D.3 MEMORY CONSUMPTION

Table 3 contains the memory consumption statistics for different network and training configurations. We can draw a number of conclusions. First, activation recomputation drastically reduces the memory consumption for pipelined backpropagation. Note that we do not observe a reduction in throughput with recomputation, as the operations to read and write stored activations also take

| Network | Local Batch Size | # IPUs | Method | Recomputation | Memory per IPU | |
| | | | | | Average | Max |
|---------|------------------|--------|--------|---------------|---------|-----|
| ResNet34 | 32 | 4 | Backprop | Y | 581.2 | 786.0 |
| | | | Backprop | N | 1221.0 | 2523.0 |
| | | | Local | N | 565.0 | 788.0 |
| | | 8 | Backprop | Y | 480.1 | 748.0 |
| | | | Local | N | 443.5 | 676.0 |
| ResNet50 | 16 | 4 | Backprop | Y | 640.2 | 787.0 |
| | | | Backprop | N | 1419.2 | 2859.0 |
| | | | Local | N | 593.0 | 785.0 |
| | | 8 | Backprop | Y | 514.1 | 819.0 |
| | | | Local | N | 447.4 | 579.0 |
| ResNet101 | 4 | 8 | Backprop | Y | 260.8 | 442.0 |
| | | | Local | N | 341.2 | 365.0 |
| | 8 | | Backprop | Y | 360.5 | 501.0 |
| | | | Local | N | 390.6 | 491.0 |

Table 3: Memory consumption in MB, for ResNets training on ImageNet with pipelined backprop and local parallelism. "Average" denotes the mean memory over IPUs, "Max" is the value for the IPU which consumed most memory. "Recomputation" refers to activation recomputation. For pipelined backprop, the number of microbatches over which gradients were accumulated was in all cases $2 \times$ the number of IPUs.

significant numbers of cycles, when not recomputing[1]. Thus, we are confident that the comparison presented in Table 1 remains valid. Even with recomputation, we see that local parallelism generally reduces the average memory consumption, and corresponds in all cases to lower or similar max memory.

The difference in memory consumption depends on the local batch size and number of processors. While recomputation reduces the number of activations that must be stored for pipelined backprop, we must still store the input to each processor for each "live" microbatch. Thus for larger microbatches this overhead increases. Further, a larger number of processors mean that there are more live microbatches at any one time, also increasing the memory overhead. Conversely, local parallelism introduces extra parameters in the auxiliary classifiers, which are not needed for backpropagation. These observations explain why the memory consumption for pipelined backpropagation increases over that of local parallelism as the local batch size and/or the number of processors grow. Thus local parallelism can reduce the memory consumption in a high throughput regime.

## E  HARDWARE BACKGROUND

The Intelligence Processor Unit (IPU) (Jia et al., 2019) is an accelerator designed for machine intelligence workloads. An IPU contains several parallel processing elements, called tiles, each of which has its own local high-speed memory (SRAM) and is able to run a number of parallel threads. For example, in the second generation IPU (MK2), each IPU chip has 1472 compute tiles; each tile is equipped with six parallel threads and 600KB of SRAM, equivalent to a total of 8832 parallel threads and 900MB of on chip memory with an aggregate 47.5 TB/s memory bandwidth per chip. This design benefits from efficient execution of fine-grained operations across the very large number of parallel threads, and allows these threads to access data efficiently. This is particularly advantageous when the data access patterns are irregular, sparse or incoherent.

---

[1]For example, for ResNet50, batchsize $4 \times 8$ over 4 IPUs we observe that pipelined backprop with recomputation has 1.5% higher throughput than without recomputation.

