# OpenReview forum: "Parallel Training of Deep Networks with Local Updates"
_ICLR.cc/2021/Conference — Reject_

### Official Review · AnonReviewer3 · 2020-10-26
**Submission aims to demonstrate how local parallelism can help improve training performance. Local parallelism refers to not having to the traditional backprop where one waits for full forward pass triggered weight update before next iteration can begin.**

**Rating:** 3
**Confidence:** 5

**Review:**

It is a very poorly written paper. Basic idea of finding a way to not have to wait for full forward pass is not new. Multiple research papers have been published from the extreme of using stale weight to some form of sub-network backdrop as a proxy for the full network. This paper proposed no new idea for local update. Prior work have all suffered with one or both of these two limitations: a) poor experimental framework, or b) not being able to meet the accuracy bar set by backprop. This work suffers from both.  Very poorly described experimental basis - and failing to come even close to the backprop accuracy target with any decent speedup claim. Former is my biggest concern. Section 6 starts with 'Here we show that performance gains of local parallelism can be realized on real hardware' - with near-zero description of any 'real' hardware, except a footnote on '1000 IPUs on a chip'.

Reference: https://deepai.org/publication/loco-local-contrastive-representation-learning - also cited in this details the same idea of local learning (see, Fig 1 - similar to overlapping in this work). And they do offer same benchmark Resnet-50 with ImageNet - meeting accuracy and memory saving.  Other cited work: arXiv:1901.08164 has shown the same local learning concept deliver accuracy for Resnet-152. Therefore, I was hoping this submission would go significantly beyond these, which it does not. It does add one language model benchmark, however, the model is very small (6M parameters).

Most concerning part though is its experimental framework - details are almost completely missing. No standard CPU-GPU details are mentioned where the experiments where conducted.  Only mention of experimental platform is in first part of Sec 6: "We implement the models in TensorFlow (Abadi et al., 2016) and train them across 4 or 8 IPUs" - then there is a footnote on that page describing hardware with 3-4 additional lines: "Intelligence Processing Units (IPUs) are massively parallel machine learning hardware accelerators. Each chip incorporates > 1000 processor cores" - This gives me no reason to believe that the authors experimented on any 'real hardware'. Hence, my overall low rating for this work.

---

> ### Author Response · Authors · 2020-11-18
> **Thank you for your feedback - clarifications on local vs backpop accuracy, experimental framework, and real hardware**
>
> We appreciate your feedback and thank you for the time spent reviewing our paper. We would very much like to improve readability and clarity of this work and thus would appreciate any sections, or areas you found to be  “very poorly written.” This can be addressed during the rebuttal period especially given the other reviewers thought the paper was well written. To summarize, the other main concerns are around (i) local vs backprop accuracy, (ii) experimental basis, and (iii) real hardware. We provide responses to all of the points of feedback and concerns below and have uploaded a revised manuscript with edits shown in blue:
>
> *Q1: Poor accuracy of backprop: "failing to come even close to the backprop accuracy”*
>
> A1: For the optimal performance backprop is the best method. That being said, our local methods are extremely close. The overlapping local method is within 0.04 cross entropy of the backprop for the Transformer, and the 2 chunk local is within 0.3% top-1 test accuracy of backprop for ResNet50. There is a tradeoff, however, as one uses more local methods performance usually suffers in exchange for faster training.
>
> Relation to loco, plus comparison to arXiv:1901.08164:
>
> The purpose of our work was explicitly to not introduce new local methods, but to compare performance in a principled way at different scales of compute. We do not claim any novelty over loco or arXiv:1901.08164. With regard to performance comparisons, loco numbers are not comparable as they are from unsupervised pre training whereas we perform purely supervised learning. With respect to arXiv:1901.08164, our resnet50 numbers outperform this related work by a significant margin (~3% top1) over the resnet152 numbers presented (which should perform better given the extra size). We go beyond this work by exploring how performance changes with different batch sizes, and with different methods whereas arXiv:1901.08164 only shows 2 models (2 chunk and backprop). Finally, as you pointed out, we do add a language benchmark with 6M parameters. Our appendix also has a larger, 136M parameter transformer on a different tokenization.
>
> *Q2: Experimental framework -- “No standard CPU-GPU details are mentioned”.*
>
> A2: These details are discussed in the appendix. We ran everything on 8-core TPU-v2 simulating the larger batch-sizes with gradient accumulation. This is NOT a real system, but meant to emulate one for research purposes. We later show on real hardware that the communication and memory overhead for local methods is lower than that for full backpropagation. We've added Appendices D and E to the manuscript to provide more details on the real hardware setup and execution traces.  We did not use a large scale real-hardware system due to the high costs (both monetary, and engineering) of building large scale systems.
>
> *Q3: Real hardware*
>
> A3: The first half of the paper was implemented to simulate training on much larger clusters. This was described in Appendix A. The second half of the paper includes experiments on real hardware. We apologize that this wasn’t clearer in the original text and have added some general background of the hardware in Appendix E. Further, we have included some utilization analysis, comprising breakdowns of communication and memory, as well as execution traces, to Appendix D. While these experiments are on less standard hardware, we don’t see this as a reason to write off the analysis, particularly since the main results with the Pareto optimal curves are hardware agnostic and show the worst case estimate (with respect to memory and communication overhead) of gains through local parallelism. We chose to use the hardware specified as it was the platform we had at our disposal. Please let us know if you’d like to know any additional details on our hardware setup.

---

### Official Review · AnonReviewer1 · 2020-10-28
**Interesting survey but poor performance evaluation**

**Rating:** 6
**Confidence:** 4

**Review:**

As someone not exposed to local training methods I found that the survey was thorough, well written and easy to understand. However, I might be missing something but it seems that the "chunked" strategy is just the same as Greedy with J divided by 2. This should probably be clarified in the manuscript.

I found section 5 particularly interesting and happened to answer most of the question I would be asking myself if I was trying to understand the behavior of these optimization methods.

My main criticism lies in the performance evaluation. Any running time analysis is highly dependent on the hardware used. First, the hardware used for this evaluation is clearly niche. It is almost impossible to get a sense of what the running times mean here. Before making any conclusion about performance I think one should compare on multiple kind of hardware or at least the most popular one. Secondly, As the authors empathize it, the drawback of training large networks is communication between machines. The fact that this setting has been overlooked in the performance evaluation is disappointing.

As the paper rightfully note, some methods might lead to poor generalization. It might be also better to use a validation accuracy cutoff instead. I also find the plots for high cutoff values uninteresting. I don't think most people care about how long it takes/how much resources are needed to train a model to a fraction of its potential.

---

> ### Author Response · Authors · 2020-11-18
> **Thank you for your feedback - clarifications on running time analysis, communication overhead, and additional details**
>
> We thank you for your feedback and time spent reviewing our paper, and are glad that you found that the “survey was thorough, well written and easy to understand.” To summarize, the main concerns are around (i) performance evaluation and (ii) hardware used. We’ve uploaded a revised manuscript with changes in blue to incorporate your feedback, and provide responses below:
>
> *Q1: “Any running time analysis is highly dependent on the hardware used”  “one should compare on multiple kind of hardware or at least the most popular one”*
>
> A1: For half of our work, we used a hardware agnostic approach based on flops (see appendix A). To justify this -- in particular, justifying ignoring memory and communication costs, we perform a series of experiments on real hardware showing throughput, and have added more experiments around memory utilization and communication (see Appendix D). In all cases, these results point to the fact that these local methods are more efficient than backprop in ways not accounted for by purely counting FLOPS. We opted to run these experiments on IPUs since they were most accessible to our lab, in terms of cost and engineering. The machine we used for this work contains high-bandwidth interconnects between processors which reduce the communication cost in distributed training setups. However, we believe that our findings will translate to other hardware platforms as the compute requirements do not change much between pipeline backprop and local methods -- local methods mostly just use less communication.
>
>
> *Q2: “As the authors empathize it, the drawback of training large networks is communication between machines. The fact that this setting has been overlooked in the performance evaluation is disappointing.”*
>
> A2: We agree with the reviewer’s observation, and hope that our recent changes address the issue.
>
> Forward and backward locking prevent efficient backpropagation over multiple devices. Pipelining mitigates this to some extent, however forward and backward communication necessitate the ramp-up and ramp-down phases in synchronous pipelined backpropagation. Utilization during these phases is poor, as we now illustrate in Appendix D. While this overhead can be reduced by accumulating gradients over a larger number of microbatches, this “eats into” the global batch size and reduces the potential for data parallelism. Local parallelism does not suffer from such locking.
>
> Through detailed profiling, we have measured the data communicated between processors. Our measurements confirm that the total data communicated with local parallelism is half that of pipelined backprop. We refer the reviewer to Appendix D for further details.
>
> *Q3: “ It might be also better to use a validation accuracy cutoff instead.”*
>
> A3: We do set the cutoff value based on validation loss on the 2 larger experiments (ResNet, Transformer). We do not include validation cutoff for the MLP model on CIFAR10 as this model is not designed to generalize and thus is less interesting from that perspective. We treat theseCIFAR10 MLP only as a proof of principle, and to test model capacity in Appendix C.
>
>
> *Q4: “I don't think most people care about how long it takes/how much resources are needed to train a model to a fraction of its potential”*
>
> A4: We agree that for some applications, lower cutoffs are not desirable. This is why we also include higher cutoff values in all of our experiments. We believe they reveal more information about how these methods work. As to your point about only caring about models that reach their full potential -- there is a recent trend in understanding optimal model size with regard to compute resources. It has been shown that it is best to use an oversized model and not train it to full capacity (https://arxiv.org/abs/2001.08361). We provide takeaways for both cases. Greedy local parallelism should be used if speed and compute-efficiency are the primary objectives. Chunked local parallelism, which matches full backprop performance, should be used if performance is the primary objective.
>
> *Q5: “"chunked" strategy is just the same as Greedy with J divided by 2”:*
>
> A5: This is correct and we will clarify the paper. More generally, chunked refers to J divided by some integer N.

---

### Official Review · AnonReviewer2 · 2020-10-28
**Good work, accept**

**Rating:** 9
**Confidence:** 3

**Review:**

The paper explores and compares several methods for parallel training of deep nets. It presents the results on multiple datasets for image classification and language modelling.

# Quality

This work provides many experiments with neat visualizations.

Pros:
- The paper provides many rigorous experiments for 7 different methods of parallel training
- The paper conducts experiments on various datasets, including image classification on CIFAR-10, language modelling on LM1B, image classification on ImageNet
- The paper conducts experiments analyzing the properties of the various methods of parallel training

Cons:
- Not many architectures explored: only Resnet for ImageNet, only Transformer for LM1B
- The paper doesn't touch on the recurrent architectures
- It is always great to see even more experiments on more datasets and tasks

# Clarity

The paper is well written and easy to follow. Although I am not an expert in the parallel training, it was easy to understand.

Pros:

- The literature is reviewed well
- The figures are compact, dense and informative

Cons:

- Sometimes, the figures are hard to read in print
- The conclusion is not very informative, I expected to see clear "do"s and "don't"s of parallel training

# Originality

The research presented in the paper is original.

# Significance

The work presented is clearly of great significance to the community.

# Conclusion

This is a thorough exploration of several methods for parallel training. The work provides a multitude of experiments comparing the given architectures. While the work doesn't introduce any groundbreaking ideas, it conducts rigorous experiments and presents them well.

Suggestions for improvements:
-  I expected to see a list of "do"s and "don't"s in the conclusion
- It is hard to read the figures when printed

# EDIT: Update

Thank you for the rebuttal.

Despite the skepticism of the other reviewers, I still think that this is a valuable, thorough paper of high quality.

---

> ### Author Response · Authors · 2020-11-18
> **Thank you for your feedback - revised conclusion to be more informative**
>
> Thank you for the valuable feedback. We are happy that you found the paper to be “well written and easy to follow” and that it “provides many rigorous experiments.” We’ve uploaded a revised manuscript with changes in blue to incorporate your feedback.
>
> Q1: Not many architectures explored
> A1: We explore 3 architectures each of which with different sizes: (4x MLP, 2x ResNet, 2x Transformer) across 2 modalities (vision, language). Exploring recurrent architectures is interesting as these have been shown to have a relatively small critical batchsize (https://arxiv.org/abs/1811.03600) or found that batch size had no effect on solution quality (https://arxiv.org/abs/1508.02788). We leave more architectures, as well as other domains (e.g. RL) for future work.
>
> Q2: conclusion is not very informative
> A2: Thank you for the feedback. We’ve summarized the main takeaways of our paper into the following points and have changed the text to make the conclusion clearer:
> 1. We are the first to extensively benchmark local parallelism with respect to performance and compute efficiency, and discuss the tradeoffs across the different methods. Greedy local parallelism should be used if speed and compute-efficiency are the primary objectives. Chunked local parallelism should be used if performance is the primary objective.
> 2. We find that local parallelism provides gains when large batch sizes are used. The takeaway is that local parallelism can be useful to prolong compute-efficient scaling, and therefore faster training, with larger batch sizes once data parallelism begins to saturate.
> 3. We are the first to show across multiple modalities (vision, language) and architectures (MLP, Resnet, Transformer).

---

### Official Review · AnonReviewer5 · 2020-11-06
**The work explores and compares different local training schemes and demonstrates the benefits of local training over synchronous backprop  on real hardware.**

**Rating:** 4
**Confidence:** 3

**Review:**

This work explores different local training schemes and demonstrates a better compute efficiency of local training schemes over the traditional synchronous backprop. The paper is well written and clearly articulates a contribution to the literature. It is intuitive and clear that local training could have advantages over backprop as it fully parallelizes the blocks of layers in DNNs. The experimental evidence is provided for both language and image classification tasks using an analytical model and real hardware. Most of the related works are cited.

Concerns:
1) Pareto optimal curves: The authors use Pareto optimal curves to show the tradeoff between total computational cost and walltime (number of sequential FLOPs) using different training schemes. Although this simplistic model provides a sense of how fast a training method could be and is agnostic to underlined hardware, the accuracy of this model is questionable as it does not consider communication and memory overhead. The roofline model used in Google TPU paper [1], which considers the memory throughput, might be a better way to estimate the walltime. Also, I think it is crucial to justify the Pareto optimal curves with real hardware experiments under a similar setting.

2) Experiments on real hardware: The current evaluation on real hardware seems to be weak as the experiments are with different local batch size and backdrop batch size. For example, I would expect local training of a deeper network to have a higher speedup than backprop. However, as the batch size is chosen differently, it is hard to compare the results and obtain insights from the results. I would appreciate a more comprehensive evaluation on real hardware.

3) Novelty: Although this paper provides an analytical model and real hardware evaluation for local training, the novelty of the paper is moderate as it does not offer new solutions or improvements over the existing local training method. As the paper mainly demonstrates the effectiveness of local training from a practical point of view, providing new insights or methodologies for implementing the local training could increase the strength of the paper.

Minor:
1) What is the number of blocks used in the experiments? There would also be a tradeoff between the number of blocks, the walltime, and accuracy. Besides, how are the layers been grouped into blocks? Does each block contain the same number of layers or a similar amount of FLOPs?
2) In Table 1, could you please provide the actual walltime (in ms) for each experiment? Without knowing the actual walltime, it is hard to know how realistic this evaluation is.
3) How does local training compared to async backprop?

[1] https://dl.acm.org/doi/10.1145/3140659.3080246

---

> ### Author Response · Authors · 2020-11-18
> **thank you - clarifications on pareto optimal curves, real-hardware experiments, and new insights / methodologies**
>
> Thank you for your valuable feedback and time spent reviewing our paper. We are encouraged that you found our paper to be well-written, and that the main point of the paper - that local parallelism enabled compute-efficiency gains over synchronous backprop - was clear. The main concerns were around (i) the format of Pareto optimal curves, (ii) real-hardware experiments, (iii) new insights or methodologies. We’ve uploaded a revised manuscript with changes in blue to incorporate your feedback.
>
> *Q1: Pareto optimal curves do “not consider communication and memory overhead”*
>
> A1: We have added tables for communication (Table 2) and memory (Table 3) overheads for local updates vs backprop, as well as discussion in Appendix D. We observe that for a given local batchsize, local methods use half as much communication, and generally less memory (depending on the batchsize and number of processors). In this respect, the Pareto optimal curves, which do not take communication or memory costs into account, are showing the worst case estimate for performance of the local methods with respect to memory & communication. We’ve clarified this point in the paper by adding Appendix D.
>
> *Q2: “crucial to justify the Pareto optimal curves with real hardware experiments”*
>
> A2: While it is true that real hardware Pareto curves would strengthen the paper, at this stage in our research we are trying to demonstrate proof of principle before scaling, as scaling is incredibly expensive both in terms of engineering requirements, as well as the price of running these large distributed systems. We do, however, show real hardware experiments measuring the throughput, performance, and communication costs of local methods vs backpropagation, and find that local methods are able to reach similar performance to backprop with higher throughput and lower communication costs.
>
> *Q3: Real hardware experiments have different local / backprop batch size. “evaluation on real hardware seems to be weak as the experiments are with different local batch size and backdrop batch size”*
>
> A3: We have intentionally used the same local batch size for local and pipeline parallelism. Keeping the global batch the same would imply a reduced local batch size, which may result in lower hardware utilization and reduced throughput for pipeline parallelism (the baseline method).
>
> Indeed, for ResNet34, 4 IPUs, using the same global batch size 32 for local and pipeline parallelism (local parallelism batch size 32, pipelined backprop microbatch size 4, number of microbatches 8), we see that local parallelism has 63% greater throughput than pipelined backprop. For ResNet50, 4 IPUs, using a global batch size 16 (local parallelism batch size 16, backprop microbatch size 2, number of microbatches 8) the factor increases to 190%. We will include these figures in Table 1 if the reviewers would like us to.
>
> *Q4: Novelty / New insights or methodologies “providing new insights or methodologies for implementing the local training could increase the strength of the paper”*
>
> A4: It is correct that we do not implement a new algorithm in this work. Instead, we opt to understand, and explore scaling of an existing, and under-explored algorithm family. To this end we provide the following insights in the paper:
>
> 1. We are the first to extensively benchmark local parallelism with respect to performance and compute efficiency, and discuss the tradeoffs across the different methods. Greedy local parallelism should be used if speed and compute-efficiency are the primary objectives. Chunked local parallelism should be used if performance is the primary objective.
> 2. We find that local parallelism provides gains when large batch sizes are used. The takeaway is that local parallelism can be useful to prolong compute-efficient scaling, and therefore faster training, with larger batch sizes once data parallelism begins to saturate.
> 3. We are the first to show results across multiple modalities (vision, language) and architectures (MLP, Resnet, Transformer).
>
> *Q5: “What is the number of blocks used in the experiments? There would also be a tradeoff between the number of blocks, the walltime, and accuracy. Besides, how are the layers been grouped into blocks? Does each block contain the same number of layers or a similar amount of FLOPs?”*
>
> A5: These details are included in the appendix. We will update the text to include a reference. For each experiment we use 1 “layer” per block. In the case of ResNet this is one residual layer (treating the input conv as a separate layer). In the case of transformers we treat each transformer layer as a separate block. For the overlapping, 2, and 3 chunk experiments, we combine more than one of these blocks together.
>
> In our experiments, each layer contains roughly a similar number of FLOPs but not identical. When building infrastructure at scale, we expect work would need to be done to balance the compute resources per chunk used for data-parallel training.

---

### Decision · Program_Chairs · 2021-01-07
**Final Decision**

**Decision:**

Reject

**Comment:**

After reading the paper, reviews and authors’ feedback. The meta-reviewer agrees with the reviewers that the paper touches an important topic(scale up training). However, as some of the reviewers pointed out, the paper could be further improved by clarifying the novelty and more thorough evaluation justification of the metric being used. Therefore this paper is rejected.

Thank you for submitting the paper to ICLR.